# Internal radiation exposure from $^{137}$Cs and its association with the dietary habits of residents from areas affected by the Chernobyl nuclear accident, Ukraine: 2016–2018

Yesbol Sartayev[1,2]*, Mutsumi Matsuu-Matsuyama[2], Izumi Yamaguchi[3], Jumpei Takahashi[4], Alexander Gutevich[5], Naomi Hayashida[1,2]*

1 Life Sciences and Radiation Research, Nagasaki University Graduate School of Biomedical Sciences, Nagasaki, Japan, 2 Division of Strategic Collaborative Research, Atomic Bomb Disease Institute, Nagasaki, Japan, 3 Ueno Hospital, Fukuoka, Japan, 4 Center for International Collaborative Research, Nagasaki University, Nagasaki, Japan, 5 Zhytomyr Inter-Area Medical Diagnostic Center, Korosten, Ukraine

* naomin@nagasaki-u.ac.jp (NH); yesbol.sartayev@gmail.com (YS)

## Abstract

The total annual effective dose has steadily decreased since the fallout of the Chernobyl Nuclear Power Plant. However, chronic internal exposure to $^{137}$Cs still persists and fluctuates in a complex and unpredictable manner. Recently, body contamination was found to primarily occur owing to the intake of forest foodstuffs that contain long-lived $^{137}$Cs. Forest foodstuffs may have up to 100 times higher concentration of cesium than does local milk and meat. The present study aimed to investigate the recent dietary habits of residents in the Zhytomyr region of Ukraine, and assess the effect of the intake of forest foodstuffs on the increase in internal radioactivity from $^{137}$Cs. We screened 1,612 participants, from July 2016 to February 2018 for internal radioactivity, using whole-body counter at Korosten Medical Center and surveyed their background and intake habits. We analyzed the association among food type, intake frequency, and internal exposure dose. The analysis revealed that nearly 90% of the participants regularly consumed one of the forest foodstuffs (mushrooms, berries, fish) or milk. Nearly 80% of the participants indicated that they consumed mushrooms or berries or both. Internal radioactivity was detected in 30% of the participants. The diet that included mushrooms exhibited the highest internal radioactivity. The lowest Bq/kg concentration was observed in the only-berry group, following the no-intake group. There was a significant correlation between the intake frequency and the magnitude of Bq/kg. Radioactivity detected in the mushroom-berry and only-mushroom group were 8.6 and 9.2 Bq/kg, respectively. The lowest and highest intake frequency showed a radioactivity of 2.4 and 7.5 Bq/kg, respectively. Radioactivity in the winter season was significantly higher than that in other seasons. In conclusion, our study revealed that internal radioactivity varies depending on the type of food, intake frequency, and season.

**Data Availability Statement:** All relevant data are within the paper and its Supporting Information files.

**Funding:** This work was supported by the Program of the Network-type Joint Usage/Research Center for Radiation Disaster Medical Science (G.A) (URL: https://housai.hiroshima-u.ac.jp/en/) and the Atomic Bomb Disease Institute of Nagasaki University (URL: https://www.genken.nagasaki-u.ac.jp/abdi/index.html)(I.Y, Y.S). The funders had no role in study design, data collection and analysis, decision to publish, or preparation of the manuscript.

**Competing interests:** The authors have declared that no competing interests exist.

## Introduction

After the accident at the Chernobyl Nuclear Power Plant (CNPP) in 1986, large amounts of radioactive materials were released from the core reactor and deposited in the soil and water surrounding the CNPP. Radioactive materials released into the air, water, and soil have raised concerns regarding internal radiation exposure and the long-term risk of cancer in individuals living in radiation-contaminated areas [1]. Chronic internal radiation exposure accounts for a substantial fraction of cumulative long-term radiation exposure among residents in radiation-contaminated areas [2]. Radiation in the human body accumulates from the ingestion of contaminated food, mostly originating from affected forests that contain considerable amounts of radioactive material absorbed from the soil [3].

Several types of radioactive materials are released from the CNPP, such as Pu, $^{90}$Sr, $^{241}$Am, $^{95}$Nb, and $^{136}$Cs, but in smaller amounts. According to an International Atomic Energy Agency report, $^{90}$Sr could cause problems in areas close to the reactor; otherwise, deposition levels were low, and radionuclides, such as plutonium and $^{241}$Am, did not cause real problems in agriculture, both because of low deposition levels and poor availability for root uptake from the soil [4]. Doses owing to $^{90}$Sr intake are relatively low [5]. $^{131}$I, $^{134}$Cs, and $^{137}$Cs are the most important radionuclides released because they are responsible for most of the radiation dose received by members of the general population [6]. $^{131}$I is short-lived, with a half-life of 8 days, but it is more critical and hazardous to the human body for several weeks immediately after the accident, especially in children aged <18 years. In total, 10–30% of iodine entered into the body predominantly accumulates in the thyroid and continue to locally expose to β-particles and γ-rays [7].

However, epidemiological studies on the consequences of the Chernobyl accident and its effects on health have reported a considerably small number of casualties than it was expected shortly after the accident [8,9]. To date, the only traceable direct health effect of radiation has been an increase in the incidence of thyroid cancer in children, a few years after the accident [8]. Children's thyroids are significantly sensitive to iodine, which accumulates in the thyroid when children inhale or consume locally produced food, milk, and meat containing $^{131}$I, immediately after an accident. In total, 20,000 thyroid cancer cases were diagnosed among highly contaminated children aged <18 years at the time of the accident; 15 of them had lethal outcomes [10].

In contrast, $^{137}$Cs is an important contributor to human radiation doses from a long-term perspective because of its prolonged half-life [11]. The half-lives of $^{137}$Cs and $^{134}$Cs are 30.2 and 2 years, respectively. Cesium has been sustained in the environment for several decades, and has chronically exposed those living in contaminated areas internally and externally for decades. Each time $^{137}$Cs is ingested, it remains in the body for approximately 100 days, which is the biological half-life of $^{137}$Cs [2]. If additional amount of $^{137}$Cs is ingested, the extra amount of radioactive particles pile up in the body in addition to the existing $^{137}$Cs, thus exposing it to $^{137}$Cs for a longer period.

Residents around the CNPP are still exposed to chronic $^{137}$Cs internal irradiation, probably because of the daily consumption of contaminated domestic foods; however, the risk of disease due to irradiation is relatively low [12]. The internal exposure dose in the body of affected individuals, except in rare cases, has been estimated to be below the International Commission on Radiation Protection (ICRP) dose limit recommendation. Therefore, the probability of stochastic outcomes would be significantly low [13]. According to ICRP 60, the estimated increased lifetime risks for fatal and nonfatal cancers from a 1 mSv/year (ICRP dose limit) $^{137}$Cs whole-body dose are 0.0005% and 0.0006%, respectively [13]. Hence, relatively little is known about the non-cancer health effects of chronic low-dose $^{137}$Cs exposure, and only a few

studies have been published on this subject [14]. Despite the risk of chronic low-dose internal exposure, most individuals living in contaminated areas continue to consume wild-forest foodstuffs for economic and habitual reasons.

This situation is most likely a great public health concern, as it continues to affect individuals, although several decades have passed since the accident. Immediately after the accident at the CNPP, to prevent internal contamination, state authorities implemented several preventive measures, such as radiation inspection of food before shipment, public education, and individual dose monitoring [3]. Nevertheless, a large proportion of individuals are exposed to internal radiation in CNPP-affected areas [15–17]. Likhtarev et al. reported that one of the reasons was the loss of fear of radiation, which led to decreased self-limitation on the use of local food products, and the other factor was the decreased ability of the government to implement countermeasures, especially in the rural areas of Ukraine, due to the deterioration of the economy. Consequently, the diet of the rural population increasingly includes more accessible and less expensive food products, produced locally or gathered in the forests [18].

However, the health effects of chronic internal exposure to $^{137}$Cs remain controversial. There is no precedent for the Chernobyl accident; therefore, long-term follow-up of $^{137}$Cs internal body burden around this area is necessary. Generally, the dose attributed to the ingestion of $^{137}$Cs in CNPP-affected regions is considered low; however, the exposure is ongoing and chronic [19]. Studies on body burden in the late 1990s and the early 2000s, primarily conducted in contaminated parts of Ukraine and Russia, have shown that the proportion of individuals with detectable internal cesium varies from one-half to one-third of the examined participants [15–20].

Local products and wild-forest foodstuffs are the main sources of internal contamination [20]. However, wild-forest products, such as berries, mushrooms, and animal meat, contain considerable amounts of $^{137}$Cs, up to 100 times higher concentration of cesium than local and home-grown meat. They contribute considerably (10–70%) to the internal effective dose [21,22]. In the domain of internal exposure, intake habits of individuals and the type of food consumed in the affected areas are the most important factors determining the level of internal exposure. Therefore, in this study, we evaluated the effects of contaminated food intake on the level of internal exposure, in the bodies of individuals living in the Zhytomyr region of Ukraine. We identified the intake frequency of forest foodstuffs among individuals in the Zhytomyr region more than three decades after the accident, and estimated the extent to which the types and frequency of intake of wild-forest foodstuffs may affect the increase in the internal radiation dose of the body.

## Materials and methods

### Study design and population

This study was conducted from July 2016 to February 2018 at the Zhytomyr Inter-Area Medical Diagnostic Center (medical center) in Korosten City. The study area is located west of the CNPP and includes Korosten City and eight other districts under the authority of the Zhytomyr region of Ukraine. The closest and farthest settlements included in our study were situated approximately 40 and 150 km west of a nuclear power plant, respectively. All participants resided within the research area at the time of participating in this study. The entire population residing in the research area was approximately 323,000 as of January 1, 2019. Korosten City was the most populous settlement in our study, with >63,000 individuals.

All patients who visited and sought medical assistance at the medical center were invited to participate in this study. During the study period, those who agreed and provided consent to participate in this study underwent whole-body counter (WBC) screening. The internal

exposure of the participants was measured by WBC manufactured by Aloka Co., Ltd. (Japan). It is a chair-type WBC with an inbuilt 7.6-cm-diameter NaI (TI) detector, where the examinee is supposed to place his/her abdomen. The WBC has an attached seat that allows the examinee to adjust his/her height and angle. The WBC was set to detect only $^{137}$Cs emitted radiation that is >270 Bq per body. All the WBC measurements were performed by qualified and competent medical personnel. All measured values of internal exposure derived from the WBC were first handwritten on hard copies of the registry cards by medical specialists.

## Data collection

We converted all individual exposure values of becquerels per body into becquerels per kilogram (Bq/kg), which is a weight-normalized body burden. Bq/kg was used to calculate the internal effective dose, using the ICRP dose conversion coefficient for $^{137}$Cs (0.0025 mSv/y/Bq/kg) [23]. When the internal exposure level of participant was below detectable level, he/she was classified as "0 Bq" and included into the "no radioactivity detected" group.

The participants completed a prescribed questionnaire aimed at identifying their dietary habits of consuming contaminated foodstuffs, frequency of consuming contaminated foodstuffs, and types of wild-forest products consumed. The questionnaires were prepared in Russian and answered by handwriting the hard copies. Respondents' answers on the intake of contaminated foodstuff frequency were inconsistent, with some specified as times per month, some per season, and some per week. Consumption frequency stratification was achieved by converting and grouping the participants based on the number of times per week. Ultimately, we grouped all of them into 4 frequency groups, such as 0, <1, <3, and ≤7 times per week. We grouped the products into main four products: berries, mushrooms, mixed mushrooms and berries, and mixed fish and milk. Some respondents specified up to four or five types of products, but in those cases, we grouped them based on the main products, such as berries, mushrooms, mixed mushrooms and berries, and mixed milk and fish. This grouping was based on foodstuff contribution to internal dose, according to previous studies [24]. For instance, if the respondent specified only mushroom, we included the respondent in the mushroom group, and if only berries then into the berry group; however, for mixtures of mushrooms, berries, fish, and milk, we included the participant in the mushroom/berry group and so forth.

All answers to the questionnaire and WBC measurement values were transferred to Excel. All available data were used to assess the effects of various variables on the internal exposure, their association, and contribution. We stratified the data according to sex, age, presence of internal contamination, type of food consumed, frequency, and season and analyzed them statistically.

All the measurements and questionnaires necessary for the study were administered at the Zhytomyr Medical Center. The medical center was the main provider of healthcare services for all residents in the study area.

## Statistical analysis

Data are presented as the averages and standard deviations for age and $^{137}$Cs Bq/kg, microsieverts (μSv), and confidence intervals for $^{137}$Cs Bq/kg. All statistical analyses were performed using IBM SPSS Statistics version 26.0 software. For statistical significance determination of average and proportions, the T-test, Mann–Whitney U test, analysis of variance (ANOVA), and chi-squared tests were used. We also ran a correlation test and univariate regression analysis to test the contributions of several variables. P-values <0.05 were considered significant.

## Ethics statements

This study was approved by the Ethics Committee of Nagasaki University Graduate School of Biomedical Sciences (approval number: 16062493, –2, –3, –4) on March 31, 2021. Written informed consent was obtained from each individual for participation in the research. Authors did not have access to information that could identify individual participants during or after data collection.

## Inclusivity in global research

Additional information regarding the ethical, cultural, and scientific considerations, specific to inclusivity in global research is included in the (S1 Questionnaire).

## Results

In total, 1,612 adult participants participated in this study. Table 1 presents the general information on the entire study population and shows the value of each item for men and women separately. Women comprised the majority of the study population (64%), and the remaining 36% were men. The average age of the entire population was 49±16.2 years. However, men were significantly younger (46±16.2 years old) than women (51±16.0 years old) (p<0.005). The average internal radioactivity of $^{137}$Cs in women was lower than that in men, accounting for 5.9 Bq/kg and 6.7 Bq/kg respectively, although the difference was not significant (p = 0.182). The average $^{137}$Cs μSv showed the same pattern as that of Bq/kg, with women exhibiting lower values than did men. $^{137}$Cs was detected in 30% of the examined population. The proportion of men who had detectable radioactivity was slightly higher (32.9%) than that of women (28.5%) (p = 0.067). The consumption of wild-forest foodstuffs did not differ significantly between men (87.3%) and women (86.8%) in 87% of the entire population indicated (p = 0.761).

The proportion and number of participants who consumed forest foodstuffs are shown in Fig 1. Fig 1A shows that 208 (13%) individuals specified that they did not consume any type of forest food and milk. The remaining 87% indicated that they consumed some type of forest foodstuff. When the participants were divided into the $^{137}$Cs detected and non-detected groups, the detected group showed a higher proportion of those who consumed forest foodstuffs, accounting for 94% (Fig 1B). Despite the $^{137}$Cs non-detected group also having a remarkably large proportion of participants consuming forest foodstuff (84%), the non-intake group was comparably larger (Fig 1C). The chi-squared test for the difference in the proportions of forest food intake between the $^{137}$Cs detected and non-detected groups was statistically significant (p<0.005).

**Table 1. General information of the participants.**

| Items | Total | Men | Women | P-value |
|---|---|---|---|---|
| Number (%) | 1,612 | 581 (36) | 1,031 (64) | |
| Age (SD) | 49 (16.2) | 46 (16.2) | 51 (16.0) | <0.001 |
| $^{137}$Cs Bq/kg (SD) | 6.2 (11.8) | 6.7 (10.9) | 5.9 (12.2) | 0.182 |
| $^{137}$Cs μSv (SD) | 15 (29) | 17 (27) | 15 (31) | 0.182 |
| Number of participants with detected $^{137}$Cs (%) | 485 (30) | 191 (32.9) | 294 (28.5) | 0.067 |
| Number of participants consuming forest foodstuffs (%) | 1,404 (87) | 508 (87.3) | 896 (86.8) | 0.761 |

The p-value column represents the significance of the difference between the values for men and women for each item.

SD, standard deviation.

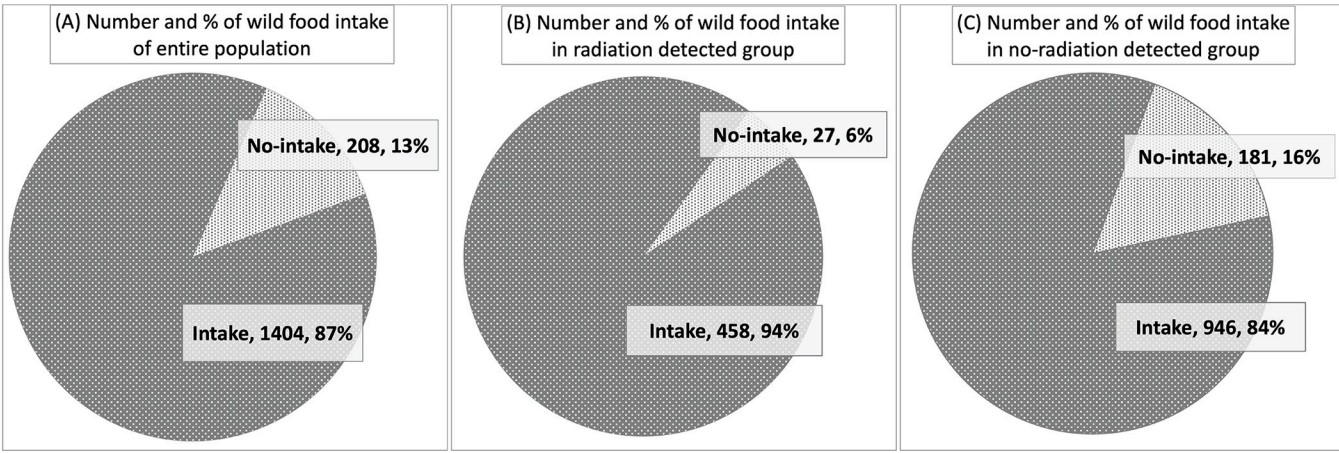

**Fig 1. Fraction of intake and no intake of forest foodstuffs and milk in each group.** (A) shows the fraction of individuals consuming forest foodstuffs and milk among all study population, with (B) showing the fractions from the radiation detected group and (C) from the no-radiation detected group. Each pie shows the fraction and number of individuals consuming and not consuming forest food, which suggests that the radiation detected group has significantly larger fraction of those who ingested forest food and milk.

Table 2 provides the results of comparison between the intake and no-intake groups. The average age of the intake and no-intake groups was 49 years (p = 0.654). The average Bq/kg of the intake group (6.8 Bq/kg) was statistically higher than that of the no-intake group (2.4 Bq/kg) (p<0.001). The average $^{137}$Cs μSv had the same pattern, with statistical significance (p<0.001). The forest foodstuff intake group had higher proportion of individuals with detected $^{137}$Cs, accounting for 32.6%, whereas the no-intake group had only 13%. The chi-squared test showed a significant difference in the proportions of detected cesium in the intake and no-intake groups (p<0.001).

We also reviewed the types of food consumed by the participants (Fig 2). The most largely consumed foodstuff was the mixed group of mushrooms and berries (39%). The average internal radioactivity of the mushroom and berry mixed group was the highest (9.2 Bq/kg) of all food types. There was a significant difference between the mushroom and berry groups and the no-intake (p<0.001), berry (p<0.001), and fish/milk (p = 0.022) groups. The second most consumed foodstuff was berries (37%), although the internal radioactivity of this intake group was the lowest of all the examined foodstuffs, measured on an average at a level of 4.1 Bq/kg. The participants who consumed only mushrooms comprised 6% of the total population, but they had the second highest average internal radioactivity (8.6 Bq/kg). Mixed milk and fish were the least consumed, accounting for only 5% of the participants, although their internal radioactivity was measured at a level of 5.0 Bq/kg on an average.

**Table 2. Comparison of the forest food intake and no-intake groups.**

| Items | Intake | No intake | P-value |
|---|---|---|---|
| Number | 1404 | 208 | |
| Age (SD) | 49 (16.0) | 49 (17.1) | 0.654 |
| $^{137}$Cs Bq/kg (SD) | 6.8 (12.3) | 2.4 (6.4) | <0.001 |
| $^{137}$Cs μSv (SD) | 17 (30.7) | 6 (16.0) | <0.001 |
| Number of individuals with $^{137}$Cs detected (%) | 458 (32.6) | 27 (13.0) | <0.001 |

The p-values in the table represent the significance of the differences between the intake and no-intake groups.
SD, standard deviation.

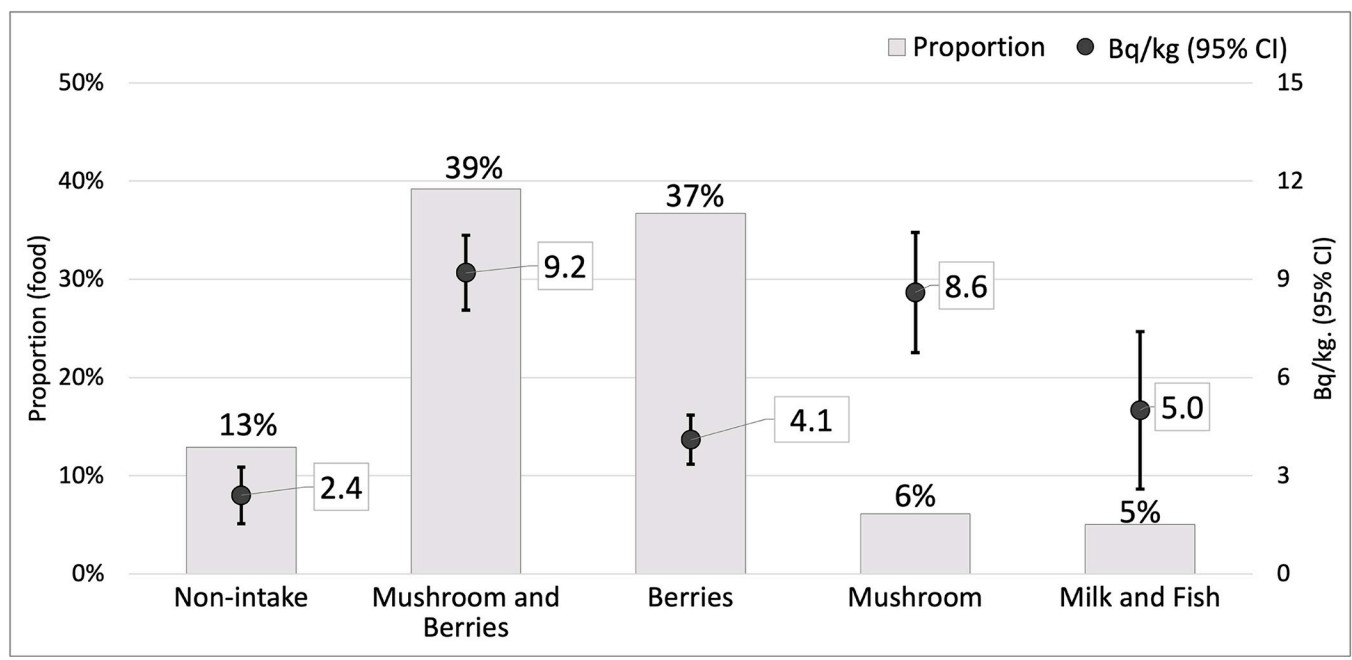

**Fig 2. $^{137}$Cs concentration (Bq/kg) and fraction based on the type of food ingested.** Bars represent the fraction of participants in each food category. The dots with whiskers represent the average internal Bq/kg and 95% confidence interval of participants in each category of food. Participants that included mushroom in their diet showed significantly higher average concentration of $^{137}$Cs. *P-value<0.05 compared with mushroom and berry category.

The frequency of intake of wild-forest foodstuffs, shown in Fig 3, is represented in times per week. The majority (43%) of all participants consumed forest foodstuffs ≥1 time or <3 times per week and had the second highest contamination in the body at an average of 6.8 Bq/kg. The second largest group, comprising 38% of the participants, consumed >0 and <1 time a week, with an average internal exposure of 6.6 Bq/kg. Only 6% of the participants indicated that they consumed forest products >3 but ≤7 times a week, with an average internal exposure of 7.5 Bq/kg. The average contamination dose increased with an increase in intake frequency. Pearson's correlation test revealed a significant positive correlation (p<0.005). The average Bq/kg of all frequency groups was significantly higher than that of the no-intake group (p<0.01).

Additionally, we performed a seasonal analysis to assess whether seasonal changes affect internal exposure (Fig 4). Winter had the highest average Bq/kg, accounting for 9.2 Bq/kg, which was significantly higher than that in the other seasons (p<0.01). Autumn showed 5.3 Bq/kg, which was significantly higher than that in the summer (p<0.01). Internal radioactivity was low in summer and spring, accounting for 2.4 Bq/kg and 3.7 Bq/kg, respectively. The ANOVA Tamhane test revealed that winter had significantly higher $^{137}$Cs concentration than all other seasons (p<0.01). The proportion of individuals with detected cesium was the highest in winter, accounting for 48.5% of the population. The proportion of individuals with detected radiation in each season was in the same order as the average Bq/kg, with the lowest proportion of individuals with detected cesium and average Bq/kg in the summer.

## Discussion

The continuous detection of chronic low-level internal contamination in individuals from affected areas after the nuclear disaster at the CNPP remains a great public health concern. In this study, although it has been more than 35 years since the accident, 30% of the participants

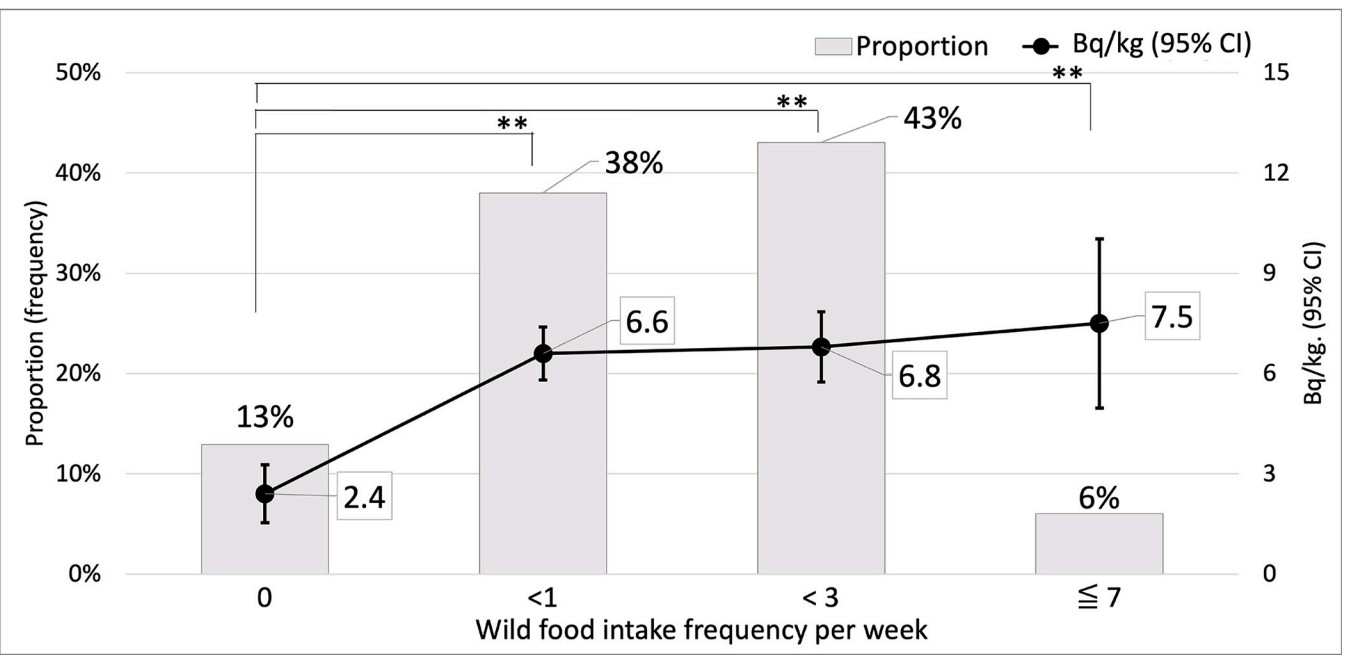

**Fig 3. $^{137}$Cs concentration (Bq/kg) and fraction based on intake frequency.** Bars represent the fraction of participants in each intake frequency (per week) group. The dots with whiskers represent the average internal Bq/kg and 95% confidence interval for each intake frequency group. The no-intake group has a significantly lower radioactivity than all the other groups. The concentration of average Bq/kg of $^{137}$Cs is elevated by the increase of intake frequency. **P-value<0.01 compared with the no-intake group (Bq/kg).

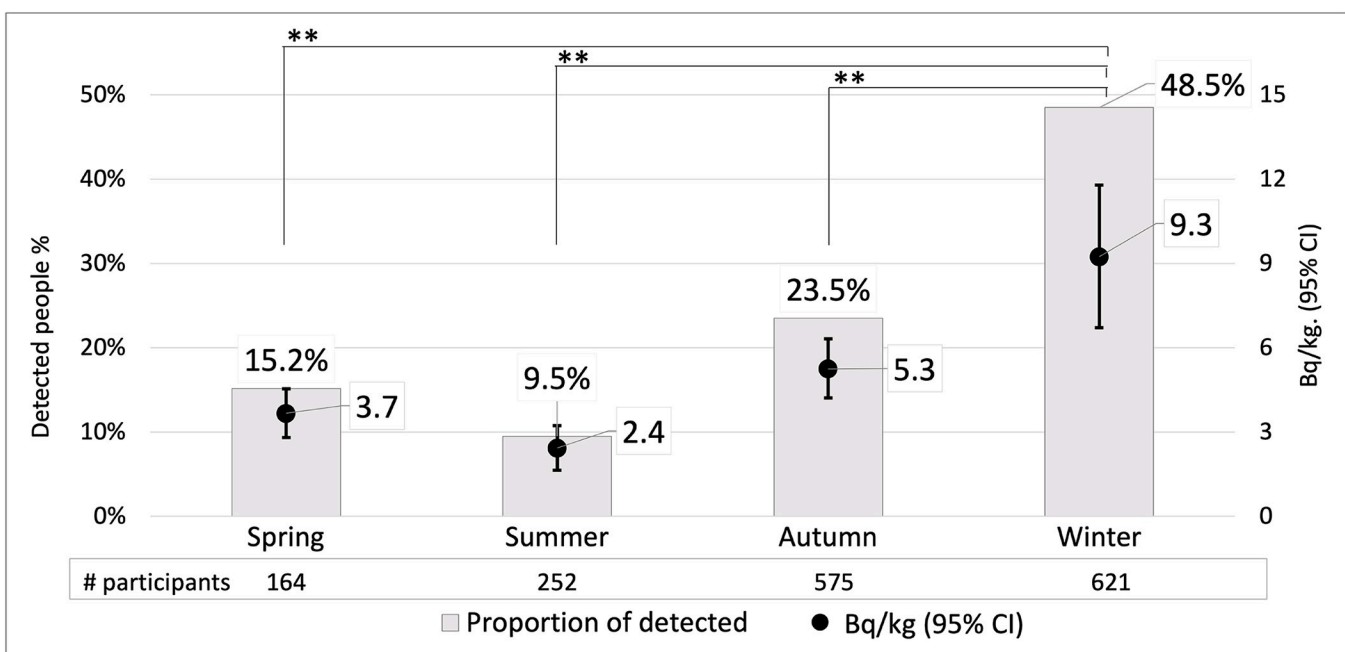

**Fig 4. Seasonal $^{137}$Cs concentration (Bq/kg) and proportion of individuals with detected radioactivity.** Bars represent the proportion of participants with detected radioactivity in each season. The dots with whiskers represent the average internal Bq/kg and 95% confidence interval in the participants of each season. Winter had significantly higher $^{137}$Cs concentration compared with that of all other seasons and largest proportion of those who exhibited internal radioactivity. ** p-value<0.01 compared to winter Bq/kg.

had internal radioactivity higher than the minimum detectable level of $^{137}$Cs. We also revealed that most of the individuals in the CNPP-affected areas consumed forest foodstuffs regularly. Nearly 90% of the respondents indicated that their diet included some type of forest foodstuff or local milk that may potentially contain cesium, although the types of forest foodstuff and intake frequency largely varied among the participants.

Internal exposure is gradually becoming more important and accounts for a substantial fraction of cumulative radiation exposure in contaminated areas. Mushrooms, forest berries, wild animal meat, and local fish are the main contributors to the internal concentration of radiation. The dose attributed to the ingestion of $^{137}$Cs in CNPP-affected regions is considerably low; however, exposure is ongoing and chronic [5]. Internal exposure in the body varies in a more complex and unpredictable manner and does not follow the temporal trend of external exposure because it fluctuates according to the dietary habits of individuals [19]. Daily lifestyle choices in radiation-contaminated areas determine internal contamination levels. It is difficult to recognize whether the detected internal contamination in participants comes from chronic intake or from one-time ingestion of contaminated food shortly before the WBC measurement. Some individuals unknowingly consume contaminated foodstuffs and incur high levels of internal contamination, whereas others consume them without hesitation.

Dietary habits that affect the concentration of radionuclides can be aggravated or intensified by poor or deteriorating socioeconomic situations because individuals in affected areas are forced to rely on contaminated forest products [25]. These products include highly $^{137}$Cs contaminated mushrooms, forest berries, wild animal meat, and fish [3,26]. Another major factor is the abundance of forest food (mushrooms, berries, games), which can have up to 100 times higher concentrations of cesium than local milk and meat and hence contributes considerably (10–70%) to the internal effective dose [21,22]. In a study by Bernhardsson in 1998, the internal exposure of participants was four times higher than that in the preceding year, presumably owing to abundance [19].

In our study, the average internal exposure in the intake group was almost threefold higher, by almost three times, than that in the no-intake group. Although we assumed that no participants in the no-intake group had detectable internal radioactivity, some participants showed internal radioactivity that was higher than the detectable level. Certainly, some respondents provided inaccurate or untruthful answers to the questionnaires. However, in some cases, cesium may have been obtained from other food products. Therefore, we assumed that the proportion of individuals who consume forest products is greater than that represented in this study. This discrepancy should be examined in future studies by administering more detailed questionnaires or through personal interviews to explore these controversies.

We observed notable differences in the consumption of mushrooms and berries between men and women. The proportion of men consuming mushrooms was nearly two times higher than women. In contrast, the proportion of women consuming berries was higher than that of men. We have previously reported that the average internal body burden was higher in men than that in women [16]. Men showed higher levels of radioactivity than did women, which may be explained by the characteristics of cesium, which tends to accumulate in muscles and bones. Thus, the dose in individuals differs depending on body size and shape [27]. However, according to our results, one of the reasons why men had higher average internal radioactivity than did women is possibly because men prefer mushrooms more than women do; mushrooms contain considerably higher concentrations of radioactivity than do berries [28].

The food that demonstrated the highest effect on the average internal exposure was the mixed group of mushrooms and berries, which were ingested by most participants. In our study, the group that consumed mushrooms alone exhibited the second highest average internal exposure. A significant difference in the average Bq/kg compared with other groups was

found in the mushroom-only and mixed mushroom and berry groups; reportedly, mushrooms have a greater effect than do other products [21,22]. The contribution of mushrooms to the mixed mushroom and berry group may be critical, as the average body burden of the berry-alone group was the lowest among all products.

We revealed a distinct tendency, suggesting that the frequency of intake of forest foodstuffs has a significant effect on the increase in the internal concentration of radioactivity. The biological half-life of $^{137}$Cs is approximately 100 days, and if contaminated food is ingested regularly, cesium accumulates over time, resulting in a higher body burden [29]. Therefore, the dietary habits of individuals living in contaminated areas are critical. Our finding that nearly 90% of the participants admitted regular ingestion of forest food indicates that individuals in contaminated areas are not concerned about contamination and consume forest products and milk without hesitation. Individuals tend to forget the past and, after a long period, become less vigilant and return to their pre-accident life habits [30]. Additionally, several decades after the accident, changes in generation may also potentially influence the attitudes of communities, although this should be investigated in future studies using appropriate methods and approaches.

We evaluated the seasonal differences in forest food intake. Wild mushrooms are consumed during a defined 3–4-month period in early summer and autumn, when fungal growth is optimal [31]. Autumn is the most effective season in terms of the magnitude of seasonal internal exposure. However, some regions tend to consume wild berries year-round [25]. In our study, winter was the most effective season, with the highest average internal exposure and highest number of individuals with detectable radioactivity. Autumn was the second most common cause of internal exposure and had the second highest number of individuals with detectable radioactivity. We found that a relatively larger proportion of participants ingested mushrooms in autumn and winter than in spring and summer. Most individuals from rural areas of Ukraine commonly consume bottled forest foods, dry berries, and vegetables during non-vegetative periods of the year, such as late autumn and winter. Handl et al. investigated the intake habits of residents living in northern Ukraine and found that dried berries and mushrooms contained remarkably higher levels of $^{137}$Cs [32]. Another study conducted in northern Ukraine concluded that increased consumption of natural products from contaminated forests, largely contributed to the increase in $^{137}$Cs concentrations in autumn and winter among residents living in highly contaminated areas and relatively less contaminated areas, even 30 years after the accident [33].

We also examined the individual that outstood from other participants with the highest internal exposure, measured with 237 Bq/kg. According to answers to the questionnaire, this female individual had consumed berries and mushrooms 3 times a week and had gotten measurement on WBC in winter. Considering the results of our research, this participant belongs to the groups that were identified with highest level of internal contamination. Most of those who consumed mushroom and berries 3 times a week were measured with the highest level of radioactivity in body. Moreover, people who have been measured in winter, showed the highest level of body contamination. Therefore, most likely this participant had consumed highly contaminated forest foodstuff, either dried or bottled, just few days before measurement on the WBC.

This study has some limitations. First, participant selection may have been biased because this study included only those who sought medical assistance from a medical center. Nevertheless, we believe that the study participants represented ordinary residents of the region with similar attitudes and intake habits. Second, the study period was relatively short compared with that of several other studies. However, as we investigated the current intake habits of forest foodstuffs and milk and their effects on the internal radioactivity concentration, we believe

that this period is sufficient for this type of study. Third, our questionnaire did not include personal interviews that may have revealed more precise results; several other forest foods that may contain $^{137}$Cs, such as wild animal meat; and the reasons that tempt individuals to consume forest foodstuffs. Adding this feature to the study methods in future studies may provide more valuable findings and progressive insights. Finally, we did not measure the concentration of $^{137}$Cs in the food consumed by the study participants or report food specifications in terms of radioactivity. The effect of each separate food on the body was not examined in depth because the diet of several participants included a mixture of various types of food. Radioactivity in the soil was not considered in this study; however, we were able to evaluate the association between food intake and internal radioactivity in the residents.

In conclusion, our results suggest that a higher intake frequency leads to higher internal exposure. Moreover, mushrooms may have the highest effect on body burden, and winter and autumn have a significantly higher average body burden. Some participants in the non-intake group may have had contamination from another food or undeliberate intake of contaminated foodstuffs, as some of them have internal radioactivity higher than the minimum detectable level in their bodies. Several individuals living in CNPP-affected areas continue to consume forest foodstuffs and milk more frequently and seem to be less concerned about contamination. Despite the continuous decrease in body burden, it varies unpredictably over time and depends on several factors that need to be studied more precisely in the future. Seemingly, the long period since the accident makes individuals feel less concerned about radiation and less reluctant about the consumption of forest foodstuffs and milk. The factors and reasons affecting internal radioactivity are crucial, and preventive measures or recommendations should be developed to avoid unnecessary excessive chronic low-dose internal exposure.

## Supporting information

**S1 Dataset.**
(XLSX)

**S1 Questionnaire. Inclusivity in global research checklist.**
(DOCX)

## Acknowledgments

The authors thank all staff members of the Zhytomyr Inter-Area Medical Diagnostic Center and all the individuals who participated in this study. We would also like to thank Editage (www.editage.com) for English language editing.

## Author Contributions

**Conceptualization:** Yesbol Sartayev, Izumi Yamaguchi, Jumpei Takahashi, Naomi Hayashida.

**Data curation:** Izumi Yamaguchi, Jumpei Takahashi, Alexander Gutevich, Naomi Hayashida.

**Formal analysis:** Yesbol Sartayev.

**Funding acquisition:** Yesbol Sartayev, Izumi Yamaguchi, Alexander Gutevich.

**Investigation:** Yesbol Sartayev, Izumi Yamaguchi, Naomi Hayashida.

**Methodology:** Yesbol Sartayev, Izumi Yamaguchi, Jumpei Takahashi, Alexander Gutevich, Naomi Hayashida.

**Project administration:** Naomi Hayashida.

**Resources:** Jumpei Takahashi, Alexander Gutevich.

**Software:** Naomi Hayashida.

**Supervision:** Mutsumi Matsuu-Matsuyama, Alexander Gutevich, Naomi Hayashida.

**Validation:** Mutsumi Matsuu-Matsuyama, Jumpei Takahashi, Naomi Hayashida.

**Visualization:** Yesbol Sartayev, Mutsumi Matsuu-Matsuyama.

**Writing – original draft:** Yesbol Sartayev.

**Writing – review & editing:** Yesbol Sartayev, Mutsumi Matsuu-Matsuyama, Naomi Hayashida.

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
