## [Decision Letter · Decision Letter 0]

27 Jul 2023

PONE-D-23-21136Internal radiation exposure from 137Cs and its association with the dietary habits of residents from areas affected by the Chernobyl nuclear accident, Ukraine: 2016–2018PLOS ONE

Dear Dr. Hayashida,

Thank you for submitting your manuscript to PLOS ONE. After careful consideration, we feel that it has merit but does not fully meet PLOS ONE’s publication criteria as it currently stands. Therefore, we invite you to submit a revised version of the manuscript that addresses the points raised during the review process.

We look forward to receiving your revised manuscript.

Kind regards,

Tim A. Mousseau

Academic Editor

PLOS ONE

Journal Requirements:

Additional Editor comments:

Thank you for submitting your manuscript to PLOS ONE.  This really is an important topic, not just for Ukraine and Belarus, but also for other parts of the world dealing with environmental contaminants in the food chain.   After careful consideration, we feel that it has merit and will be acceptable for publication following a few minor changes. Therefore, we invite you to submit a revised version of the manuscript that addresses the points raised during the review process.

Please carefully review the two reviewers' comments and make appropriate revisions to your manuscript. The main issues include greater detail concerning the whole body counting system used and the possible role of alpha emitters that are rarely measured but could be important. Please provide a comprehensive description of the WBC system. As for alpha emitters, I am not sure how much information really exists for Chernobyl populations but please  try to address this question if possible. 

Reviewers' comments:

Reviewer's Responses to Questions

**Comments to the Author**

1. Is the manuscript technically sound, and do the data support the conclusions?

Reviewer #1: Partly

Reviewer #2: Partly

2. Has the statistical analysis been performed appropriately and rigorously? 

Reviewer #1: Yes

Reviewer #2: Yes

3. Have the authors made all data underlying the findings in their manuscript fully available?

Reviewer #1: Yes

Reviewer #2: Yes

4. Is the manuscript presented in an intelligible fashion and written in standard English?

Reviewer #1: Yes

Reviewer #2: Yes

5. Review Comments to the Author

Reviewer #1: The authors have conducted a large-scale survey of internal radiation exposure among the inhabitants of a regional city (Zhytomyr) in Ukraine. I generally agree with the content of the manuscript, but I am convinced that the manuscript would be better if the points listed below were revised. I look forward to your consideration of the modifications.

1.

The Abstract only presents qualitative research methods, results and conclusions. At the very least, the years of investigation and the key conclusions should be "quantitatively" described.

2.

I have some questions about the detectable limits. The WBC used in the study is described as a type with a 7.6 cm (3-inch, i think) NaI detector, Aloka, but it maybe probably a chair type. If this is the case, please indicate specifically how the detection limit (or MDA) of 270 Bq/body was calculated, using the 3sigma or Currie's method.

3.

Supplemental information indicates that there is individual with extremely high radioactivity levels of 237 Bq/kg (29 years old, see attaached image file). Statistics are important, but discussing this individual case is crucial to understanding the reality of the exposure. I would like to know the reasons why authors did not mentioned about this case.

4.

L150 modify "0.0025 msv/y/Bq/kg" to "0.0025 mSv/y/Bq/kg"

Reviewer #2: The manuscript "Internal radiation exposure from 137Cs and its association with the dietary habits of residents from areas affected by the Chernobyl nuclear accident, Ukraine: 2016–2018" is interested, well written and presents part of the problem connected to radiation from the Chernobyl accident.

Such screening studies are important, especially when internal exposure, both ingestion and inhalation, is the main source of effective dose. However, the authors have focused on beta emitters only. In the first few days iodine was the biggest problem, indeed; but during the accident extremely high amount s of other nuclides were released, including alpha-emitters which ionize the body in much bigger level. Thus, the information about iodine or Cs-134 (Introduction) seems to be less important. In my opinion the authors must add the information about alpha emitters. At the moment the amount of Cs-137 is two times smaller, while most of alpha emitters are still present in the environment in unchanged quantity. It is hard to measure them but they cause the highest ionizing effect in the body.

Anyway, the studies were well planned and organized, the analysed group is quite impressive, the analysis to the foodstuff types (the main sources of Cs-137) was done and presented clearly. The figures should be improved, their quality is insufficient. After small remarks the manuscript is worth publishing in PLOS One.

6. PLOS authors have the option to publish the peer review history of their article (what does this mean?). If published, this will include your full peer review and any attached files.

Reviewer #1: No

Reviewer #2: No

---

## [Author Response · Author response to Decision Letter 0]

29 Aug 2023

Editor

We appreciate your earnest comments and valuable suggestions that have notably reinforced the structure and improved the comprehension of our manuscript. We have considered all the comments and suggestions that were submitted and responded respectively as provided below.

Comment 1. Please ensure that your manuscript meets PLOS ONE’s style requirements, including those for file naming.

Response. We made sure that the manuscript meets all of PLOS ONE’s style requirements and as well as the file naming. (Links for templates 1 and 2)

Comment 2. Please include a complete copy of PLOS’ questionnaire on inclusivity in global research in your revised manuscript. Our policy for research in this area aims to improve transparency in the reporting of research performed outside of researchers’ own country or community. The policy applies to researchers who have travelled to a different country to conduct research, research with Indigenous populations or their lands, and research on cultural artefacts. The questionnaire can also be requested at the journal’s discretion for any other submissions, even if these conditions are not met. Please find more information on the policy and a link to download a blank copy of the questionnaire here: https://journals.plos.org/plosone/s/best-practices-in-research-reporting. Please upload a completed version of your questionnaire as Supporting Information when you resubmit your manuscript.

Response. We completed the questionnaire on inclusivity and uploaded with our revised manuscript under the title of file “S1 Questionnaire”. 

Comment 3. Please review your reference list to ensure that it is complete and correct. If you have cited papers that have been retracted, please include the rationale for doing so in the manuscript text or remove these references and replace them with relevant current references. Any changes to the reference list should be mentioned in the rebuttal letter that accompanies your revised manuscript. If you need to cite a retracted article, indicate the article’s retracted status in the References list and also include a citation and full reference for the retraction notice.

Response. We have ensured that the reference list is complete and correct. We did not find record of retraction of any paper we listed in the reference that are part of our manuscript. 

Reviewer #1:

We are grateful to reviewer #1 for the critical comments and useful suggestions that have helped us to improve our paper considerably. As indicated in the responses that follow, we have taken all these comments and suggestions into account in the revised version of our paper.

Comment 1. The Abstract only presents qualitative research methods, results and conclusions. At the very least, the years of investigation and the key conclusions should be "quantitatively" described.

Response. We have revised the content of the abstract and added quantitative information regarding the research details and conclusion. 

Comment 2. I have some questions about the detectable limits. The WBC used in the study is described as a type with a 7.6 cm (3-inch, i think) NaI detector, Aloka, but it maybe probably a chair type. If this is the case, please indicate specifically how the detection limit (or MDA) of 270 Bq/body was calculated, using the 3sigma or Currie's method.

Response. The Aloka Whole Body Counter (WBC) is a chair-type equipment. We modified the sentence describing the WBC, so the reader can have a better understanding. The minimum detectable activity of WBC is set by the producer of the equipment. Therefore, we did not adjust MDA nor calculated it. 

Comment 3. Supplemental information indicates that there is individual with extremely high radioactivity levels of 237 Bq/kg (29 years old, see attaached image file). Statistics are important, but discussing this individual case is crucial to understanding the reality of the exposure. I would like to know the reasons why authors did not mention about this case. 

Response. Our research was designed mainly to focus on the general population; thus, we primarily examined all participants as groups that are represented on average. However, we agree that some exceptional cases should be observed and discussed in detail. We incorporated a paragraph about this particular case that was pointed out by the reviewer, 

in the discussion part of our manuscript, 

Comment 4. L150 modify "0.0025 msv/y/Bq/kg" to "0.0025 mSv/y/Bq/kg".

Response. We modified the description above as suggested.

Reviewer #2:

We are grateful to reviewer #2 for the critical comments and useful suggestions that have helped us to improve our paper considerably. As indicated in the responses that follow, we have taken all these comments and suggestions into account in the revised version of our paper.

Comment 1. The authors have focused on beta emitters only. In the first few days iodine was the biggest problem, indeed; but during the accident extremely high amounts of other nuclides were released, including alpha-emitters which ionize the body in much bigger level. Thus, the information about iodine or Cs-134 (Introduction) seems to be less important. In my opinion the authors must add the information about alpha emitters. 

Response. We agree that iodine and Cs-134 are less important in terms of ionizing effect on the body compared to that of alpha-emitters; however, our intention by mentioning them was to provide an overall picture of radioactive materials released from CNPP, particularly taking into account the amount of major radionuclides released. Also, this additional information was provided for those who have limited understanding about radioactive materials and their effects. 

Comment 2. At the moment, the amount of Cs-137 is two times smaller, while most of alpha emitters are still present in the environment in unchanged quantity. It is hard to measure them, but they cause the highest ionizing effect in the body. 

Response. We admit the fact that alpha emitters are still present in the environment, probably in unchanged quantity. However, the main goal of our research was Cs-137 and its chronic effect on the body of those living in contaminated areas. Even now, after several decades, Cs-137 persists in the environment in larger amounts despite its ionizing effect being halved owing to half-life longevity. Cs-137 is still being detected in people from contaminated areas, and it is also easily measurable and traceable. 

Comment 2. The studies were well planned and organized, the analysed group is quite impressive, the analysis to the foodstuff types (the main sources of Cs-137) was done and presented clearly. The figures should be improved, their quality is insufficient. After small remarks the manuscript is worth publishing in PLOS One.

Response.

We appreciate all the suggestions listed above and will certainly take into consideration in our future research projects. We made some minor modifications to our figures, as suggested.

---

## [Editor Report · Decision Letter 1]

31 Aug 2023

Internal radiation exposure from 137Cs and its association with the dietary habits of residents from areas affected by the Chernobyl nuclear accident, Ukraine: 2016–2018

PONE-D-23-21136R1

Dear Dr. Hayashida,

We’re pleased to inform you that your manuscript has been judged scientifically suitable for publication and will be formally accepted for publication once it meets all outstanding technical requirements. Congratulations, and thank you for submitting this important study to PLoS ONE.

Kind regards,

Tim A. Mousseau

Academic Editor

PLOS ONE
---

## [Editor Report · Acceptance letter]

7 Sep 2023

PONE-D-23-21136R1 

Internal radiation exposure from 137Cs and its association with the dietary habits of residents from areas affected by the Chernobyl nuclear accident, Ukraine: 2016–2018 

Dear Dr. Hayashida:

I'm pleased to inform you that your manuscript has been deemed suitable for publication in PLOS ONE. Congratulations! Your manuscript is now with our production department. 

Kind regards, 

on behalf of

Dr. Tim A. Mousseau 

Academic Editor

PLOS ONE